# Epigenetic Alterations from Barrett’s Esophagus to Esophageal Adenocarcinoma

**DOI:** 10.3390/ijms24097817

**Published:** 2023-04-25

**Authors:** Pelin Ergun, Sezgi Kipcak, Serhat Bor

**Affiliations:** 1Ege Reflux Study Group, Division of Gastroenterology, Faculty of Medicine, Ege University, 35040 Izmir, Türkiye; pelin.ergun@ege.edu.tr (P.E.); kipcaksezgi@gmail.com (S.K.); 2Department of Medical Biochemistry, Faculty of Medicine, Ege University, 35040 Izmir, Türkiye; 3Department of Medical Biology, Faculty of Medicine, Ege University, 35040 Izmir, Türkiye

**Keywords:** Barrett’s esophagus, adenocarcinoma, epigenetics, methylations, miRNA

## Abstract

Barrett’s esophagus (BE) is a disease entity that is a sequela of chronic gastroesophageal reflux disease that may result in esophageal adenocarcinoma (EAC) due to columnar epithelial dysplasia. The histological degree of dysplasia is the sole biomarker frequently utilized by clinicians. However, the cost of endoscopy and the fact that the degree of dysplasia does not progress in many patients with BE diminish the effectiveness of histological grading as a perfect biomarker. Multiple or more quantitative biomarkers are required by clinicians since early diagnosis is crucial in esophageal adenocancers, which have a high mortality rate. The presence of epigenetic factors in the early stages of this neoplastic transformation holds promise as a predictive biomarker. In this review, current studies on DNA methylations, histone modifications, and noncoding RNAs (miRNAs) that have been discovered during the progression from BE dysplasia to EAC were collated.

## 1. Introduction

BE is a columnar cell dysplasia characterized as a phenotype of gastroesophageal reflux disease (GERD) with high-grade esophagitis, which can progress to esophageal adenocarcinoma (EAC) [1]. Dr. Norman Barrett first mentioned this type of dysplasia in his 1950 article titled *“Chronic peptic ulcers of the esophagus and esophagitis”* [2], and since then, thousands of articles have been published regarding this aberrance, particularly regarding its association with esophageal cancer. A meta-analysis reported a worldwide prevalence of 3–14% for histologically confirmed BE [3]. The incidence of BE is steadily increasing in Western societies, where it is more prevalent compared to Eastern societies [4,5]. BE can lead to the development of a more severe condition known as low-grade dysplasia (LGD) or high-grade dysplasia (HGD). These abnormal cells can progress to intramucosal carcinoma and eventually become invasive carcinoma without treatment [6]. Patients with GERD are 3.1 times more likely to develop EAC compared to those without GERD. However, the likelihood of developing EAC is significantly higher in patients with BE, which is 29.8 times greater compared to those without BE [7]. The prognosis for EAC is generally poor, as approximately >50% of cases are typically diagnosed at advanced stages (III–IV). This is a major contributing factor to the low 5-year survival rate of EAC patients, which has recently been reported to range between 20.1% and 23.4% [8,9].

The objective of regular endoscopic surveillance accompanied by histopathological examination in patients with BE and EAC is to identify dysplasia or neoplasia at an early stage [10]. Therefore, the identification of specific biomarkers in the detection of LGD or HGD and EAC is important due to its potential for early intervention and cancer stage determination, including its cost-effectiveness and applicability compared to upper gastrointestinal endoscopy [11]. It has become widely acknowledged that epigenetic factors, specifically those related to super-enhancers, DNA methylation, histone modifications, and non-noncoding RNAs, can be inherited somatically and can play a role in creating lasting but adaptable alterations in the development and advancement of HGD and EAC [12,13]. Early stages of cancer development involve the initiation of epigenetic modifications, which are indicative of the likelihood of progression. There are various epigenetic modifications that occur during carcinogenesis, including DNA methylation, changes to histone proteins after translation, certain types of miRNA, and alterations to nucleosome positioning [10,14].

This review aimed to provide an overview of the current literature on biomarker research focusing on epigenetic changes and their potential role in the progression from BE to EAC, which summarizes DNA methylation, histone modifications, and noncoding RNAs that may contribute to the development of EAC in individuals with BE (Figure 1).

### 1.1. DNA Methylation as Biomarkers BE and EAC

DNA methylation is an important epigenetic modification that involves the addition of a methyl group to cytosine residues in DNA. Normal methylation patterns are necessary for cell growth and metabolism, whereas abnormal methylation can lead to diseases such as tumors [15]. Other epigenetic modifications, such as histone modifications and noncoding RNAs, also play important roles in gene regulation and development. Environmental factors can also influence epigenetic patterns, highlighting the importance of gene–environment interactions. In vertebrates, DNA methylation occurs when a methyl group binds to the CpG sites by DNA methyltransferase; as a result, five methyl cytosines are formed. Methylated cytosines are found in approximately 75% of all CpG dinucleotides in the human genome [16]. The enzyme that adds methyl groups to the CpG islands in DNA is called DNA methyltransferase (DNMT). DNA methylations can inhibit gene expression either directly or indirectly through methyl-CpG binding domain proteins, thereby suppressing protein expression [17].

In all cancer types, hypermethylation and hypomethylation are observed in DNA. Wide areas of hypomethylation are observed globally, whereas hypermethylation is observed in specific regions such as CpG islands and localized areas of hypermethylation in gene promoter regions [14]. There are significant differences in the amount and distribution of DNA methylation between different vertebrate tissues because DNA methylation varies by species and tissue, highlighting the importance of tissue-specific epigenetic regulation for proper gene expression and differentiation [18].

### 1.2. DNA Hypermethylation Is a Frequent Event in BE and EAC

DNA methylation has also been studied extensively in BE. Hypermethylation of CpG islands and hypomethylation have distinctive hallmarks in BE progression [19]. Abnormal methylation of CpG islands has been examined in BE, dysplastic BE, and EAC. The hypermethylation of cyclin-dependent kinase inhibitor 2A (*CDKN2A*), a stabilizer of the tumor suppressor protein p53 and cell cycle G1 control inhibitor [20], has been shown in studies conducted nearly 20 years ago [21,22,23,24,25,26]. The *CDKN2A* gene is located on chromosome 9p21 and has two different upstream exons (1α and 1β) regulated by different promoters. The transcript that was initiated from the proximal promoter (1α) encodes *CDKN2A–p16INK4A,* while the latter (1β) encodes *CDKN2A–p14ARF* [20]. The majority of *CDKN2A* studies in BE, HGD, and EAC is on *CDKN2A–p16INK4A* promoter methylation. This chromosome loss and hypermethylation of the promoter of the *CDKN2A–p16INK4A* inhibit the activity of the *CDKN2A* gene. Inactivation of this gene has been reported in patients with BE, dysplasia, and EAC [14,21,26,27]. Although the frequency of *CDKN2A* hypermethylation in BE mucosa ranges from 3% to 77%, it varies between 11% and 75% in dysplastic tissues and between 16% and 85% in EAC (Table 1). Furthermore, CpG island hypermethylation of the *CDKN2A* promoter was either absent or very low in the normal squamous epithelium (Table 1). These findings suggest that *CDKN2A* methylation is an early change in BE formation.

However, hypermethylation of this gene is not specific to BE, suggesting the need for multiple methylation markers. A panel including *TP53* loss of heterozygosity (LOH), *CDKN2A* LOH, and tetraploidy was screened in esophageal biopsies from 243 BE patients, and it was shown that the co-existence of these three abnormalities has an approximately 39-fold risk of cancer progression. However, it was stated that the methylation of *CDKN2A* by itself did not cause a significant statistical difference [55]. A retrospective cohort study of 50 progressive and 145 nonprogressive BE patients by Jin et al. reported that a panel of 8 biomarkers including *CDKN2A* and age factor had an area under the curve (AUC) of 0.732 at diagnosis [56]. Schulmann et al., in their retrospective cohort study with 77 EACs, 93 BEs, 20 dysplasias (*n* = 14 LGD, *n* = 6 HGD), and 64 NE patients, reported that *CDKN2A, RUNX3*, and *HPP1* inactivation and progression risk increased 2 years before EAC diagnosis in patients with BE [24]. *CDKN2A* is also regulated independently by the *p14ARF* promotor [57]. However, Vieth et al. showed that *p14ARF* promoter hypermethylation can also be observed frequently in Barrett adenocarcinoma (20%) [25]. It has been reported that during the progression from normal epithelium to BE and EAC, there is a significant decrease in *p14ARF* expression [58].

Hypermethylation of the *CDKN2A* promoter may be induced by nicotinamide adenine dinucleotide phosphate hydrogen (*NADPH*) oxidase 5 (*NOX5*) overexpression in BE and EAC. NOX5 is a novel NADPH oxidase that produces superoxide. Reactive oxygen species (ROS) produced by *NOX5* have been implicated in the immune system stimulation and signaling cascades during tumorigenesis [59]. Some cell culture studies in BE and EAC have shown that *NOX5* messenger RNA (mRNA) levels are higher than in healthy tissues [60,61]. It has been reported that NOX5-mediated accumulation of hydrogen peroxide (H_2_O_2_) suppresses its own expression and causes *CDKN2A* promoter hypermethylation due to DNMT1 upregulation [61,62].

These studies show that *CDKN2A* promoter methylation is an important marker in the progression from BE to EAC. However, these studies are mostly qualitative, which means they cannot provide clear data regarding the degree of methylation that is crucial in the prognosis. In a recent study, lesion-containing sections of patients with LGD, HGD, and EAC were separated from histological sections by laser-capture microdissection, and *p16* (encoded by *CDKN2A*) methylation analysis was performed with pyrosequencing. The authors reported that *p16* methylation increased in parallel with the level of the carcinogenesis process [40]. 

In hypermethylation panel studies, such as the studies by Schulmann et al. [24] and Jin et al. [56], Runt-related transcription factor 3 *(RUNX3)* is also found to be hypermethylated in Barrett’s dysplasia (Table 1) [24,33,56]. This tumor suppressor gene, which has a role in the transforming growth factor β signaling pathway, has been frequently deleted or transcriptionally silenced in cancer [63]. In nine studies selected in a meta-analysis examining *RUNX3* promoter methylation in the development of esophageal cancer, it was found that *RUNX3* methylation of EAC was significantly higher than in healthy controls and patients with BE. *RUNX3* promoter methylation has been reported to be an important independent risk factor for the progression of BE to HGD of the esophagus and EAC [64].

The downregulation of adenomatous polyposis coli (*APC*) hypermethylation is another topic of EAC progression. *APC* is a negative regulator of the Wnt/β-catenin pathway. The loss of *APC* expression causes the stabilization and nuclear accumulation of β-catenin, which can lead to the initiation of tumorigenesis [30]. Hypermethylation of the *APC* promoter has been studied in many cancers, including EAC (Table 1). In the combined hypermethylation panel study including the *APC* promoter, conducted by Clement et al. with 12 progressive BE and 16 nonprogressive BE patients, it was reported that the *APC* promoters of progressive patients and nonprogressive patients were 100% and 36% hypermethylated, respectively. However, no statistical analysis was performed in the study [29]. In a study examining the methylation status of *APC, CDKN2A*, mutL homolog 1 *(hMLH), RUNX3*, and Methylguanine methyltransferase (*MGMT)* genes, only *APC* gene hypermethylation was reported to be an independent predictor of EAC [65]. Wang et al. showed that p16 and *APC* promoter hypermethylation in 17 normal esophagus, 102 BE, and 42 adenocarcinoma patients is reported to be an important marker in dysplastic BE and EAC during a mean follow-up of 4.1 years [30]. In another study, it was reported that *APC* promoter hypermethylation was found in approximately 40% of BE patients and 92% of EAC patients in esophageal biopsies, whereas it was not detected in the esophagus of healthy controls. Plasma samples from the same patients were also taken, and methylated *APC* was found in 25% of the plasma of EAC patients [35].

These results suggest that aberrant methylation of this gene occurs in the early stages of the BE to EAC process. However, more precise data are warranted to determine the diagnostic power of *APC* methylation and its relationship with EAC progression. In a meta-analysis conducted by Wang et al. to investigate the early detection potential of *APC* hypermethylation in esophageal cancer, 18 studies showed that *APC* hypermethylation was higher in esophageal cancer (EC) and BE than controls in the data of 1008 ECs, 570 BEs, and 782 controls [66]. They also stated that *APC* methylation in ECs was similar to BE and not associated with tumor stage and survival. Researchers reported that the diagnostic performance of *APC* methylation shows an AUC of 0.94 in EC and 0.88 in BE.

β-catenin interacts with e-cadherin (*CDH1*), the intercellular synthesis molecule, and with APC, the tumor suppressor gene product. APC competes with e-cadherin for binding with beta-catenin [67]. E-cadherin is a tumor suppressor protein, and the downregulation of e-cadherin in tumor cells is frequently observed in metastasis [68]. In epithelial cancers, e-cadherin may be silenced by LOH in the 16q22 region, promoter hypermethylation [20], or inherited mutations such as germline large deletions [69] and transcriptional silencing [70] in gastric cancers. However, diagnostic inherited mutations in BE and EAC have not been demonstrated thus far in EC or BE. In a study with healthy controls and esophageal cancer patients, e-cadherin hypermethylation was not observed in healthy controls, whereas methylation was observed in 84% of esophageal tissues of cancer patients (Table 1) [37]. In a similar methylation panel study, 66% of patients with EAC were reported to have methylation [32]. Schildhaus et al. found that e-cadherin methylation was rare (1/10) in patients with Barrett carcinoma [38].

One of the main mechanisms triggering cancer is ROS-mediated DNA damage. ROS damages DNA, RNA, enzymes, and proteins, which are involved in the activation of oncogenes and inhibition of tumor suppressors [71]. The exposure of the esophageal epithelium to acid influences the pathogenesis of BE, and acid is considered a carcinogen [72]. Glutathione peroxidase 3 (*GPx3*) is the main scavenger of ROS in plasma and is responsible for H_2_O_2_ detoxification. *GPx3* is expressed in many tissues of the human gastrointestinal tract, including the esophageal squamous epithelium [20]. One study showed that *GPx3* mRNA expressions were reduced to approximately 90.5% in Barrett’s adenocarcinoma samples. In the same study, the authors concluded that 61.9% of BE, 81.8% of dysplastic samples, 88.2% of EAC samples, and 17% of normal squamous epithelium had *GPx3* promoter hypermethylation (Table 1) [45]. However, they discussed that the methylation in normal samples may be attributed to abnormal contaminant cells. These results show that intestinal metaplasia cells lose their ability to ROS detoxification. In addition, *GPx3* hypermethylation may be an important marker that develops in the early period of BE carcinogenesis.

As seen in Table 1, the frequency of hypermethylation in well-studied genes such as CDKN2A/p16 varies among studies. Although methylation-specific PCR is mostly used in these studies, the use of methods such as methylight [34] or melt curve analysis [33] may also contribute to this variability. Additionally, the difficulty in diagnosing and monitoring patient groups such as Barrett’s dysplasia, the lack of information on dysplasia grade in each study, and the small sample sizes may explain the differences in ranges.

### 1.3. DNA Hypomethylation in BE and EAC

In the prognosis of BE and EAC, global hypomethylation is an important distinguishing characteristic, along with hypermethylation of CpG islands. Hypomethylation levels in DNA have been linked to a higher likelihood of genome instability [73] and increased expression of oncogenes such as Cyclin D1 (*CCND1*), Cyclin E1 (*CCNE1*), *KRAS, MYC,* and cell division protein kinase 6 (*CDK6*) [74]. A study showed that global hypomethylation is present at the earliest stages of epithelial carcinogenesis in BE [75]. The study also found that epigenetic regulation during epithelial carcinogenesis may not be restricted to traditionally defined “CpG islands” but also occur through differential methylation outside these regions. Finally, they found that novel targets X-C motif chemokine ligand 1 (*XCL1*), matrix metallopeptidase 13 (*XCL3*), GATA binding protein 6 (*GATA6*), and deleted in malignant brain tumors 1 (*DMBT1*) were more highly expressed in NDBE and HGD/EAC tissues compared to normal squamous epithelium. This suggests that the observed hypomethylation may contribute to the development and progression of BE and EAC by activating genes that promote cell proliferation and survival. Another study showed that patients with BE and EAC have decreased DNA methylation levels outside the CpG islands and increased methylation in the CpG islands compared to the squamous epithelium [19].

Hypermethylation and hypomethylation lead to global changes in the transcriptome that are involved in the development of EAC and appear early in carcinogenesis. In a study examining differentially methylated CpG sites in patients with BE and EAC, there were 15 CpG sites that demonstrated differential methylation between the BE hypermethylated epigenotype and BE hypomethylated epigenotypes and 74 CpG sites in the EAC [14]. In a retrospective cohort study, methylation profiles of 150 BE and 285 EAC cases were examined and grouped into 4 subtypes: subtype 1 with aberrant DNA methylation, a high mutation rate, and multiple mutations in the cell cycle and receptor tyrosine signaling pathways; subtype 2 with a metabolic gene expression pattern and unmethylated transcription factor binding sites; subtype 3 with no changes in methylation; and subtype 4 with DNA hypomethylation, which is linked to structural changes and copy number variations, with increased amplification of *CCNE1*. Subtype 4 exhibited hypomethylation, large-scale genomic rearrangements, copy number alterations, and *CCNE1* and erb-b2 receptor tyrosine kinase 2 (*ERBB2)* amplification. The findings of this study reveal the heterogeneity of DNA methylation patterns in BE and EAC and their impact on gene expression and genomic stability, providing evidence for the involvement of DNA methylation changes in EAC development [74]. Boldrin et al. stated that LINE-1 hypomethylation could be used as a biomarker not only to monitor EAC prognosis but also as an indicator in BE surveillance [76]. In a study examining the whole genome methylation of progressive and nonprogressive nondysplastic BE patients, 44 methylation profiles were found to be different between the two groups; particularly, hypomethylation at the OR3A4 position was identified as a differentiator between progressive and nonprogressive patients [13].

Studies have demonstrated that the epigenomes of BE and EAC are frequently hypomethylated in intragenic and noncoding regions [14]. Furthermore, hypomethylation may be more dominant than hypermethylation in BE progression [56,75]. A high-resolution methylome analysis study showed that in the early stages of BE progression, there is a greater tendency toward hypomethylation rather than hypermethylation as the primary epigenetic alteration, and early progression of BE is characterized by genome-wide hypomethylation affecting both coding and noncoding regions of the genome [77].

### 1.4. Histone Modifications in BE and EAC Pathogenesis

Histones are a group of small proteins that are vital components of chromatin, the material that makes up chromosomes. They comprise four different types, known as H2A, H2B, H3, and H4. Each histone has a spherical section and a flexible charged tail, which extends from a protein complex called a nucleosome. A nucleosome is a cluster of 8 histone proteins that wrap around approximately 146 base pairs of DNA. These histone octamers are made up of two copies of each of the four core histones [78]. Chromatin is a nucleoprotein complex composed of DNA, histones, and nonhistone proteins that serves as the fundamental framework for storing eukaryotic genetic information. Regulation of gene expression is achieved through modifications to the histone tails, including acetylation, methylation, phosphorylation, ubiquitination, sumoylation, proline isomerization, and ADP ribosylation. These modifications are post-translational and allow fine-tuned control over gene expression [79]. The alterations to histone modifications can be undone and are regulated by enzymes that comprise histone acetyltransferases (HATs) and deacetylases (HDACs), as well as methyltransferases and demethylases [80]. For instance, histone acetylation is a chemical modification that influences DNA structure and gene transcription. When lysine residues are acetylated by HATs, the DNA structure relaxes, which facilitates gene transcription. Conversely, hypoacetylation of histones is a characteristic feature of inactive heterochromatin. In cancer cells, there is an impaired balance between HATs and HDACs, resulting in a significant alteration in the chromatin structure. This, in turn, leads to changes in gene expression, particularly those related to cell cycle regulation, differentiation, and apoptosis. Thus, cancer cells display aberrant gene expression patterns due to the disrupted balance between HATs and HDACs [10,81].

Despite the potential importance of histone modifications in BE and EAC, relatively few studies have been conducted in this area. Many of the studies that have been conducted have focused on esophageal squamous cell carcinoma (ESCC), a different type of esophageal cancer that is not closely related to BE or EAC [82]. The study investigated the expression of histone deacetylase 1 (*HDAC1)* and histone deacetylase 2 (*HDAC2)* genes in EAC and found that both genes were highly expressed in cancerous tissues. However, there was no correlation between *HDAC1* expression and tumor stages. In contrast, the study revealed a significant correlation between overexpression of *HDAC2* and increased lymphatic spread of the tumor, as well as aggressive tumor behavior [83].

To gain a comprehensive understanding of the role of histone modifications in the progression from BE to EAC, there is a need for further detailed and extensive investigations, similar to those that have been conducted for ESCC.

Moreover, the identification of additional histone modifications and the characterization of their roles in BE and EAC could provide further insights into the pathogenesis of these diseases and potentially lead to the development of novel therapeutic targets. In conclusion, the examination of histone modifications in BE and EAC is a promising area of research with significant implications for the diagnosis and treatment of these diseases, but further studies are needed to fully understand their roles and potential as biomarkers and therapeutic targets.

### 1.5. The Role of miRNAs as Biomarkers in BE and EAC

MicroRNAs (miRNAs) are small, noncoding RNA molecules of approximately 20–22 nucleotides that regulate gene expression by binding to target mRNAs leading to their post-transcriptional modifications. miRNAs play important roles in regulating important and disease-related cellular processes such as cell proliferation, differentiation, and apoptosis. In the context of cancer, certain microRNAs can play opposing roles in tumor development and progression. Oncomir microRNAs promote tumor growth and metastasis by downregulating tumor suppressor genes, whereas tumor suppressor microRNAs inhibit tumor growth by targeting oncogenes. These opposing actions highlight the complex role of microRNAs in cancer biology and their potential as therapeutic targets [84].

While the exact mechanisms underlying the transition from BE to EAC are not fully understood, research has shown that miRNAs might play a role in this process. Many studies have investigated miRNAs as potential biomarkers and therapeutic targets for the early detection and treatment of BE and EAC. These studies have identified specific miRNAs that are dysregulated in BE and EAC tissues and have demonstrated their potential as diagnostic and prognostic biomarkers. Despite these promising findings, miRNA-based biomarkers and therapeutics for BE and EAC have yet to be translated into clinical practice. Further research is needed to validate the diagnostic and prognostic utility of miRNAs and to develop effective miRNA-targeted therapies for these pathologies. Nevertheless, the identification of dysregulated miRNAs in BE and EAC tissues has provided valuable insights into the molecular mechanisms underlying these diseases and has opened up new avenues for their diagnosis and treatment.

miR-192 acts as a tumor suppressor by inhibiting cell proliferation and inducing apoptosis in different cancer types such as colon and prostate cancers [85,86]. Conversely, some studies have reported that miR-192 has oncogenic properties by promoting cell proliferation, migration, and invasion in different cancer types such as bladder and pancreatic cancers [87]. Studies examining the role of miR-192 in BE and EAC suggest that it generally acts as a tumor suppressor (Table 2). Hassan et al. found that miR-192 expression was significantly reduced in BE tissues compared with normal esophageal tissues. Reduced expression of miR-192 was associated with an increased risk of progression from BE to EAC. They also showed that overexpression of miR-192 in EAC cells inhibited cell proliferation and invasion, suggesting that miR-192 may function as a tumor suppressor in EAC [88]. miR-192 has the potential to be a noninvasive biomarker for the diagnosis of BE in general. Further studies are needed to confirm the findings in larger patient cohorts and to explore the clinical utility of microRNA-based diagnostic tests for BE [89,90].

miR-194 is generally considered to be a tumor suppressor miRNA. Studies have shown that miR-194 expression is reduced in many types of cancer, including gastric, colorectal, and breast cancers. This reduction in miR-194 expression is often associated with increased tumor growth, invasion, and metastasis [101,102]. miR-194 has been identified as a predictive marker for the progression of BE and EAC. The research findings indicated that miR-194 expression levels were notably elevated in tissue samples obtained from patients who had developed EAC [90,92]. A thorough investigation conducted on serum samples collected from individuals with BE and EAC revealed that, similar to the findings from tissue samples, miR-194 expression levels were markedly elevated in the serum of these patients as compared with those with normal esophageal epithelium [103]. In addition, increased miR-194 gene expression has been associated with intestinal epithelial differentiation by Hino et al. [104]. Although strong evidence postulates that miR-194 is associated with metaplasia and neoplastic progression, reaching a definite conclusion regarding whether it has oncomir or tumor suppressor properties remains unattainable [105].

The miRNAs miR-192 and miR-194 have been extensively researched and found to exhibit elevated expression levels in BE and EAC. The high miR-192 and -194 expression in BE and EAC is due to hypomethylation in the promoter regions [89].

miR-215 can act as an oncomir or as a tumor suppressor, depending on the mRNA targets. For instance, this miRNA has tumor suppressor properties in many cancer types, while in gastric cancer, miR-215 has been shown to promote malignant progression by targeting the RUNX family transcription factor 1 (*RUNX1*) gene, which is involved in cell differentiation and apoptosis [106,107]. Several studies have investigated the expression of miR-215 in BE and EAC. The results of these studies have consistently shown that mir-215 expression is higher in both BE and EAC compared to normal esophageal tissue [98,108,109]. However, when the expression levels between BE and EAC were compared, two studies found that the expression of miR-215 was lower in EAC [90,92]. This finding is particularly significant for the potential use of miR-215 as a biomarker during the transition from BE to EAC. Additionally, the decreased expression of miR-215 in EAC suggests that it may have tumor suppressor properties.

miR-203 plays a crucial role as a tumor suppressor in various types of cancer [110]. The ability to bind the 3′ UTR of mRNA of jun proto-oncogene (*c-Jun)*, which is a transcription factor involved in regulating cellular growth and differentiation, has potential in cancer treatment strategies. miR-203 can inhibit the *c-Jun* mRNA translation, which results in a reduction of c-Jun protein levels. Therefore, miR-203 is considered a promising therapeutic molecule in the treatment of a variety of cancers [111,112]. It also plays a role in suppressing the growth of tumors in EAC [109]. Research conducted by Hezova et al. has shown that EAC patients with low levels of miR-203 in their tumor tissue have a shorter period of time without disease recurrence [99]. Furthermore, studies have revealed that the expression of miR-203 tends to decrease during the progression from healthy esophageal tissue to EAC [92]. Another significant feature of miR-203 is its ability to target and downregulate tumor protein *p63* mRNA. P63 is a protein that is expressed in differentiated suprabasal cells and promotes their division [113,114]. In cancers such as the EAC, the low expression of miR-203 is believed to be due to hypermethylation of the miR-203 gene, leading to the increased expression of p63 [115]. By inhibiting the translation of p63, miR-203 prevents the growth and proliferation of cancerous cells.

miR-205 is a molecule known for its role as a tumor suppressor [96]. It has been found to be expressed at low levels in both BE and EAC tissues compared to normal epithelial tissue [88,97]. Interestingly, when comparing the expression levels, it was found that BE had relatively higher expression levels compared to EAC [100]. This finding is particularly significant for understanding the transition from BE to EAC. One important aspect of miR-205’s tumor suppressor function is its association with epithelial–mesenchymal transition (EMT). Low expression of miR-205 leads to the increased expression of its targets, zinc finger E-box binding homeobox 1 and 2 (*ZEB-1* and *ZEB-2*, respectively), which in turn decrease the expression of e-cadherin, promoting EMT [116]. This relationship between miR-205 and EMT provides compelling evidence for miR-205’s tumor suppressor properties. However, Hezova et al. conducted a study on cell lines and found that miR-205 exhibits a tumor suppressor effect in EAC by regulating the EMT pathway, while in ESCC, it has an oncogenic effect through the regulation of matrix metalloproteinase-10 [117]. Additionally, in a different study in ESCC, overexpression of miR-205 led to the development of radiation-resistant properties, promoting cancer growth [118].

Profiling BE- and EAC-specific miRNAs is a crucial area of study to understand BE to EAC transition. Saller et al. endeavored to develop a prognostic method for the progression of BE to EAC. They conducted a comparative analysis of biopsy samples from patients with BE who did not experience dysplasia or carcinoma during a 7-year follow-up period (BE-nonprogressed, BEN) and those who developed carcinoma during a 3–4-year follow-up period (BE-progressed, BEP). To profile 24 biopsy samples of BE (comprising 13 BENs and 11 BEPs), the researchers utilized the NanoString nCounter miRNA assay. They determined the most significantly differentially expressed miRNAs between the 2 groups and selected the top 12 miRNAs (miR-1278, miR-1301, miR-1304-5p, miR-517b-3p, miR-584-5p, miR-599, miR-103a-3p, miR-1197, miR-1256, miR-509-3-5p, miR-544b, and miR-802) for principal component analysis. The 12-miRNA signature demonstrated high sensitivity and specificity in both the training and validation datasets for distinguishing between BEN and BEP. Overall, the researchers were successful in identifying a 12-miRNA signature that could reliably differentiate between BEN and BEP using miRNA profiling. This discovery has the potential to enhance the prediction of BE progression and facilitate the earlier detection of esophageal adenocarcinoma [119].

The process of identifying potential biomarkers for the transition from BE to EAC measured from esophageal samples should ideally be noninvasive. Li et al. endeavored to uncover a panel of miRNAs capable of accurately distinguishing between BE and normal esophageal tissue. To achieve this, they employed a noninvasive esophageal cell sampling device called cytosponge. The authors analyzed miRNA expression profiles using the Agilent microarray and Nanostring nCounter assays from two distinct sets of esophageal biopsy tissues obtained during endoscopy from 38 BE patients and 26 controls with a normal esophagus. Their analysis revealed 15 miRNAs that were significantly upregulated in BE tissues compared to controls, of which 11 were verified in Cytosponge samples. The most prominently upregulated miRNAs in BE tissues were miR-196a, miR-192, miR-194, and miR 215, each of which exhibited an area under the curve (AUC) value of 0.82 or more for distinguishing BE from control tissues. The researchers developed an optimized multivariable logistic regression model based on the expression levels of 6 miRNAs that could identify BE patients with an AUC value of 0.89, 86.2% sensitivity, and 91.6% specificity. Furthermore, the combination of miR-192, miR-196a, miR-199a, and trefoil factor 3 expression levels showed an AUC of 0.93, 93.1% sensitivity, and 93.7% specificity in identifying patients with BE. Finally, the researchers identified a miRNA expression pattern capable of identifying Cytosponge samples from BE patients with an AUC of 0.93. The overexpression of miR194 in BE samples through epigenetic mechanisms might be implicated in the pathogenesis of BE [89].

Circulating miRNAs, as well as tissue, have high potential as biomarkers for the early detection of EAC malignancy. In a systematic screening study conducted by Wang et al. using serum samples from patients with BE and EAC, they found that miR-130a was highly expressed in BE and EAC patients. Additionally, they found that high-grade BE patients highly expressed miR-130a than low-grade BE patients. When comparing early-stage and advanced-stage EAC patients, they also found that miR-130a expression increased as the stage advanced. These results provide significant evidence that miR-130a is associated with the development of BE and EAC [93]. In a recent study published by Hassan et al., they showed that seven miRNAs had ROC curves that could distinguish between BE, EAC, and HG neoplasia. Among the miRNAs with significant expression, miR-92a-3p’s data were compared with tissue samples, and it was determined that the main source of miR-92a-3p in circulation was epithelial cells supporting neoplastic cells [94].

miRNAs can not only be tools for diagnosis or demonstrate progression but also predictive biomarkers of response to anti-cancer therapies and potential therapeutic targets. Therefore, it is crucial to identify their target genes [120]. Yao and colleagues aimed to identify the miRNA–mRNA regulatory network for distinguishing BE from EAC, and they selected 16 molecules as hub genes based on enriched function and pathway analyses. They determined that *CDH1*, phosphoribosylglycinamide formyltransferase (*GART*), G2 and S-phase expressed 1 (*GTSE1*), NIMA-related kinase 2 (*NEK2*), miR-496, miR-214, and miR-15b were associated with survival [95]. 

The association of miRNA expressions with intricate molecular networks renders it challenging to determine miRNAs that can serve as both biomarkers and therapeutic targets. As a result, extensive research in this field is necessary to explore the potential for promising advancements. 

## 2. Conclusions

BE results from chemical damage and is a significant risk factor for EAC. Regular endoscopic follow-ups and tissue biopsies are valuable tools for investigating the molecular changes associated with BE, including epigenetic modifications such as DNA methylation, histone modifications, and miRNA regulation. Notably, these epigenetic changes can occur in BE prior to the development of dysplasia or cancer. However, caution is necessary when utilizing these changes to predict disease progression since not all individuals with BE progress to dysplasia or cancer. Although the inactivation of tumor suppressor genes and activation of oncogenes is commonly observed in BE, more research is required to determine the potential of these epigenetic changes as markers for disease progression.

## Figures and Tables

**Figure 1 ijms-24-07817-f001:**
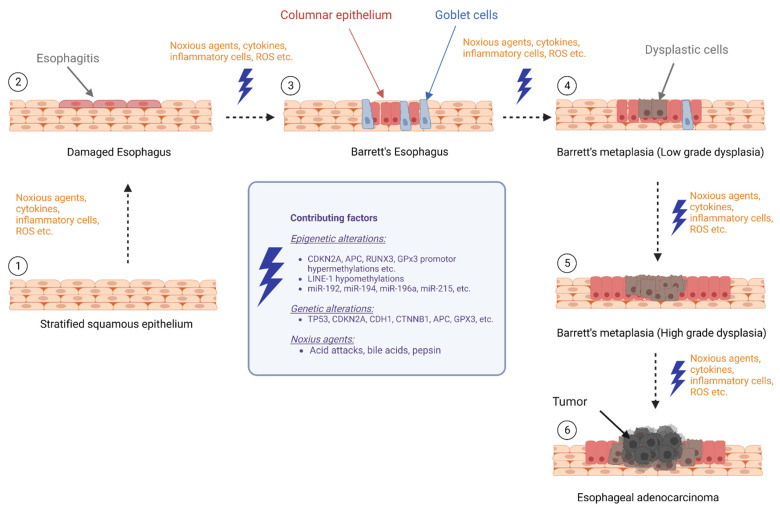
Epigenetic alterations during progression of Barrett’s esophagus to esophageal cancer (BE: Barrett’s esophagus, LGD: low-grade dysplasia, HGD: high-grade dysplasia, and EAC: esophageal adenocarcinoma) (The figure created with biorender.com, accessed on 23 March 2023).

**Table 1 ijms-24-07817-t001:** Hypermethylated genes in Barrett’s Carcinoma.

Hypermethylated Genes	Main Function	Classification	Hypermethylation Ranges	References
*AKAP12*	a cell-growth-related protein	acts as a tumor suppressor	BE (39%), BD (53%), EAC (52%) NSE (0%)	[28]
*APC*	negative regulator that controls beta-catenin	tumor suppressor	BE (36–95%, mean: 67%) [7] BD (50–89%, mean: 61%) [3] EAC (42–95%, mean: 74%) [9] (additionally, 25% in plasma of EAC) [1] NSE (0–14%, mean: 3%) [5]	[24,27,29,30,31,32,33,34,35] In plasma [35]
*CCNA1*	controls proliferative and survival activities in tumorigenesis	oncogenic	BE (81%), BD (68%), EAC (90%) NSE (1%)	[36]
*CDH1 (e-cadherin)*	an essential transmembrane protein within adherens junctions	tumor suppressor	EAC (10–84%, mean: 53%) [3] NSE (0–12%, mean: 6%) [2]	[32,37,38]
*CDH13*	a member of the calcium-dependent cell adhesion molecule family	tumor suppressor	BE (70%), BD (78%), EAC (76%) NSE (0%)	[39]
*CDKN2A*	regulates the cell cycle	tumor suppressor	p16INK4A promotor: BE (3–77%, mean: 29%) [12] BD (11–75%, mean: 40%) [9] EAC (16–85%, mean: 51%) [14] NSE (0–43%, mean: 9%) [9] p14ARF promotor: BE (8%) [1] EAC (0–20%, mean: 7%) [3] NSE (0%) [1]	p16INK4A [21,22,24,25,27,30,31,32,33,34,40,41,42] p14ARF [24,25,27]
*DAPK*	positive mediator of gamma-interferon-induced programmed cell death	tumor suppressor	BE (50%) [1], BD (53%) [1], EAC (20–70%, mean: 50%) [3] NSE (5–20%, mean: 13) [2]	[32,38,43]
*ESR1*	plays a role in growth, metabolism, sexual development, and gestation	tumor suppressor	BE (69%) [1] BD (67–100% mean: 84%) [2] EAC (51–100%, mean: 76%) [2] NSE (5–12%, mean: 9%) [2]	(Broc [31,32]
*EYA4*	possesses phosphatase, hydrolase, and transcriptional activation functions	tumor suppressor	BE (77%), EAC (83%) NSE (3%)	[44]
*FHIT*	role in the regulation of apoptosis	tumor suppressor	EAC (70%)	[38]
*GPx3*	H_2_O_2_ detoxification	acts as a tumor suppressor	BE (62–90%, mean: 76%) [2] BD (82–88%, mean: 85%) [2] EAC (62–88%, mean: 75) [2] NSE (8–17%, mean: 13%) [2]	[45,46]
*GPx7*	catalyzes the reduction of H_2_O_2_	acts as a tumor suppressor	BE (18%), BD (80%), EAC (67%) NSE (0%)	[46]
*GSTM2*	detoxification of electrophilic compounds	acts as a tumor suppressor	BE (50%), BD (56%), EAC (69%) NSE (5%)	[46]
*GSTM3*	detoxification of chemical substrates or electronic compounds	acts as a tumor suppressor	BE (13%), BD (38%), EAC (15%) NSE (0%)	[46]
*HPP1/TMEFF2*	may play multiple roles in cell growth, maturation, and adhesion	tumor suppressor	BE (44–75%, mean: 60%) [2] BD (79–100%, mean: 90%) [2] EAC (71–83%, mean: 77%) [2] NSE (3–4% mean: 4%) [2]	[24,33]
*ID4*	inhibits of DNA binding protein family	acts as a tumor suppressor	BE (78%), BD (86%), EAC (78%) NSE (21%)	[33]
*MGMT*	DNA repair enzyme that plays an important role in chemoresistance to alkylating agents	tumor suppressor	BE (25–89%, mean: 52%) [6] BD (89–100%, mean: 95%) [2] EAC (21–79%, mean: 51%) [8] NSE (17–55%, mean: 27%) [5]	[24,32,33,34,38,42,47,48]
*NELL1*	encodes a cytoplasmic protein that contains epidermal growth factor-like repeats	promising tumor suppressor	BE (42%), BD (53%), EAC (48%) NSE (0%)	[49]
*RBP1*	the carrier protein involved in the transport of retinol	tumor suppressor	BE (58%), BD (57%), EAC (70%) NSE (18%)	[33]
*RUNX3*	activate or suppress transcription, role in the TGF-β signaling pathway	tumor suppressor	BE (25–48%, mean 37%) [2] BD (57%) [2] EAC (48–73%, mean: 61%) [2] NSE (2%) [2]	[24,33]
*SFRP1*	affects cell growth, limits the cell cycle, and induces apoptosis	tumor suppressor	BE (81–100%, mean: 91%) [3] BD (100%) [1] EAC (91–95%, mean: 93%) [3] NSE (10–17%, mean: 14%) [2]	[29,33,50]
*SOCS1*	potent inhibitor of the interferon gamma (IFNγ) pathway	acts as a tumor suppressor	BE (0%), BD (13%), EAC (42%) NSE (0%)	[51]
*SOCS3*	regulates cytokine or hormone signaling	acts as a tumor suppressor	BE (13%), BD (46%), EAC (74%) NSE (0%)	[51]
*TAC1*	encodes peptides that target nerve receptors, immune cells, stem cells, hematopoietic cells, and smooth muscle cells	acts as a tumor suppressor and oncogenic	BE (56%), BD (58%), EAC (61%) NSE (8%)	[52]
*TERT*	ensuring chromosomal stability by maintaining telomere length	acts as a tumor suppressor	BE (17%), BD (92%), EAC (64%) NSE (0%)	[29]
*TIMP3*	Complexes with metalloproteinases and irreversibly inactivates them	tumor suppressor	BE (23–88%, mean: 63%) [6] BD (78%) [1] EAC (20–90%, mean: 63%) [7] NSE (0–19%, mean: 6%) [4]	[24,29,32,33,34,42,53]
*VIM*	supporting and anchoring the position of the organelles in the cytosol	Interacts with tumor suppressors	BE (91–100%, mean: 96%) [2] BD (63–100%, mean: 82) [2] EAC (77–81%, mean: 79%) [2] NSE (0–1%, mean: 1%) [2]	[36,54]

*CDKN2A*: cyclin-dependent kinase inhibitor 2A; *APC*: Adenomatous polyposis coli; *RUNX3*: Runt-related transcription factor 3; *GPx3*: Glutathione peroxidase 3; *GPx7*: Glutathione peroxidase 7; *GSTM2*: Glutathione S-transferase Mu 2; *GSTM3:* Glutathione S-transferase Mu 3; *SOCS-1*: Suppressor of cytokine signaling 1; *SOCS-3*: Suppressor of cytokine signaling 3; *VIM*: Vimentin; *CCNA1:* Cyclin-A1; *EYA4*: eyes absent 4; *ESR1*: Estrogen Receptor 1; *HPP1:* hyperpigmentation, progressive, 1; *TMEFF2*: transmembrane protein with EGF-like and two follistatin-like domains 2; *TIMP3:* Metalloproteinase inhibitor 3; *MGMT*: Methylguanine methyltransferase; *CDH13:* cadherin 13; *NELL1*: Neural EGF-like protein 1; *TAC1*: Tachykinin Precursor 1; *AKAP12*: A-kinase anchor protein 12: *DAPK*: death-associated protein kinase; *FHIT*: fragile histidine triad; *TERT*: Telomerase reverse transcriptase; *SFRP1*: Secreted frizzled-related protein 1; *RBP1*: Retinol binding protein 1; *ID4*; Inhibitor Of DNA Binding 4; NSE: Normal squamous epithelium, BD: Barrett’s dysplasia, EAC: Esophageal adenocarcinoma, [n]: number of studies.

**Table 2 ijms-24-07817-t002:** Deregulated microRNAs in BE and EAC.

Deregulated microRNAs	Classification	Sample Types	References
miR-192	tumor suppressor	BE, EAC (tissue and serum, cytosponge)	[88,91]
miR-194	oncomir or tumor suppressor properties	BE, EAC (tissue and serum, cytosponge)	[89,90]
miR-215	tumor suppressor	BE, EAC (tissue and serum, cytosponge)	[90,92]
miR-203	oncomir	BE, EAC (tissue and serum)	[89,90]
miR-205	tumor suppressor	BE, EAC (tissue and serum)	[89,90]
miR-130a	oncomir	BE, EAC (serum)	[93]
miR-92a-3p	oncomir	BE, EAC (tissue and serum)	[94]
miR-496, miR-214, miR-15b	tumor suppressor	BE, EAC (enrichment analysis *)	[95]
Let-7c	tumor suppressor	BE, EAC (tissue)	[88,96]
miR-196a	oncomir	BE, EAC (tissue and cytosponge)	[2,89,90,97]
miR-31	oncomir	BE, EAC (tissue)	[98,99,100]
miR-375	oncomir	BE, EAC (tissue)	[98]

BE: Barrett’s esophagus, EAC: Esophageal adenocarcinoma. * The results of the study are reinforced by enrichment analysis. *Enrichment analysis* is a computational approach that is commonly used in bioinformatics to identify biological pathways, cellular processes, and functions that are significantly enriched with genes or miRNAs that are differentially expressed between two or more conditions.

## Data Availability

Not applicable.

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
