# Peer review of "Epigenetic Alterations from Barrett’s Esophagus to Esophageal Adenocarcinoma"

_ijms, 2023, doi:10.3390/ijms24097817_

Round 1

Reviewer 1 Report

This manuscript summarized epigenetic mechanisms in Barrett’s esophagus and esophageal adenocarcinoma, focusing on DNA methylation and miRNAs. Generally, this summary article will provide a comprehensive information about what epigenetic events occur during Barrett’s tumorigenesis and are they useful as biomarkers? However, the manuscript was not organized well that makes the readers hard to follow and get clear conclusion about the issues discussed. And the language is not so professional and need substantial improvement. In addition, the authors should read the original publications carefully to make sure the summaries are accurate.

Some suggestions:

1.       For Table1, for each individual genes, list the prevalence (or a range) of DNA hypermethylation in normal BE, BD and EAC so readers can easily figure out if this is an early or late event?

2.       Separate the discussion about DNA methylation of each gene in BE, BD and EAC from the application as biomarker. For example, use the subtitles like: DNA hypermethylation is a frequent event in BE and EAC; DNA hypomethylation in BE and EAC; DNA methylation as biomarkers BE and EAC…… Similar strategy can be used for miRNAs too.

3.       Page 1 line 32-34, “Patients with GERD are 1.7 times more likely to develop EAC compared to those without GERD. However, the likelihood of developing EAC is significantly higher in patients with BE, which is 10.6 times greater compared to those without BE [7]”. This is not accurate based on the reference 7. The table 2 is for esophageal cancer but Table 3 is for EAC. Please read your references carefully to make sure the summary in your manuscript is correct.

4.       Figure 1 misses very important components that induce BE and EAC, Bile acids. Also lack the important stage of GERD. What are the Noxious agents, genetic and epigenetic factors? Please list the major factors under each step.

5.       Page 3 line 91, “Inactivation of this gene has been reported in patients with BE (3%–77%), dysplasia (LGD: 20%–56% and HGD: 60%–75%), and EAC (39%–85%) [19,30–32]. Furthermore, CpG island hypermethylation of the CDKN2A promoter has not been observed in the normal squamous epithelium [31–34].”  For CDKN2A gene, what were the incidences of DNA hypermethylation in BE, dysplasia and EAC?

6.       Table 1 misses many important genes such as GSTM2, GPX7, MT3, etc.

7.       Page 5 line 120, “CDKN2A has two different upstream exons (1α and 1β) regulated by different promoters. The transcript that was initiated from the proximal promoter (1α) encodes CDKN2A–p16INK4A, while the latter (1β) encodes CDKN2A–p16INK4A [29].” What is the deference between 1a and 1 β? They encode the same protein?  1βencodes CDKN2A–p14ARF?

8. Page 5 line 147, “RUNX3 is also found to be hypermethylated in Barrett’s hyperplasia”. What is Barrett’s hyperplasia? 

Author Response

Dear Reviewer 1,

Thank you for reviewing our article, "Epigenetic Alterations from Barrett’s Esophagus to Esophageal Adenocarcinoma". We appreciate your feedback, and we have revised our work according to your suggestions. We have attached the revised version of our article, along with a point-by-point response to your comments, which are highlighted in yellow. Please see the attach manuscript.

I also want to inform you that the English language editing of the article was conducted by the Ege University Planning and Monitoring Coordination of Organizational Development and Directorate of Library and Documentation. A thank you statement has been added to the acknowledgement section of the article.

We hope that these revisions address your concerns.

Thank you for your help in improving the quality of our work.

Some suggestions & The Answers

  1. For Table1, for each individual genes, list the prevalence (or a range) of DNA hypermethylation in normal BE, BD and EAC so readers can easily figure out if this is an early or late event?

Thank you for your feedback. We also believe that your suggestion has made hypermethylation more understandable. In Table 1, the methylation ranges, numbers of the studies (n), and averages for each gene methylation in normal BE, BD, and EAC were provided. Gene names were rearranged in alphabetical order. HPP1/TMEFF2 were merged as the same gene.

  1. Separate the discussion about DNA methylation of each gene in BE, BD and EAC from the application as biomarker. For example, use the subtitles like: DNA hypermethylation is a frequent event in BE and EAC; DNA hypomethylation in BE and EAC; DNA methylation as biomarkers BE and EAC…… Similar strategy can be used for miRNAs too.

The subheadings were organized as you suggested.

  1. Page 1 line 32-34, “Patients with GERD are 1.7 times more likely to develop EAC compared to those without GERD. However, the likelihood of developing EAC is significantly higher in patients with BE, which is 10.6 times greater compared to those without BE [7]”. This is not accurate based on the reference 7. The table 2 is for esophageal cancer but Table 3 is for EAC. Please read your references carefully to make sure the summary in your manuscript is correct.

Thanks for clearing up the confusion. The sentence has been corrected as follows based on Table 3: "Patients with GERD are 3.1 times more likely to develop EAC compared to those without GERD. However, the likelihood of developing EAC is significantly higher in patients with BE, which is 29.8 times greater compared to those without BE." in line 34-36.

  1. Figure 1 misses very important components that induce BE and EAC, Bile acids. Also lack the important stage of GERD. What are the Noxious agents, genetic and epigenetic factors? Please list the major factors under each step.

A chronic GERD figure depicting the transition from squamous epithelium to Barrett's esophagus has been added. We have also included genetic, epigenetic, and noxious factors that affect the progression of BE and EAC.

  1. Page 3 line 91, “Inactivation of this gene has been reported in patients with BE (3%–77%), dysplasia (LGD: 20%–56% and HGD: 60%–75%), and EAC (39%–85%) [19,30–32]. Furthermore, CpG island hypermethylation of the CDKN2A promoter has not been observed in the normal squamous epithelium [31–34].” For CDKN2A gene, what were the incidences of DNA hypermethylation in BE, dysplasia and EAC?

The line has been changed as “Inactivation of this gene has been reported in patients with BE, dysplasia and EAC [14,21,26,27]. Although the frequency of CDKN2A hypermethylation in BE mucosa ranges from 3% to 77%, it varies between 11% and 75% in dysplastic tissues and between 16% and 85% in EAC (Table 1). Furthermore, CpG island hypermethylation of the CDKN2A promoter was either absent or very low in the normal squamous epithelium (Table 1). These findings suggest that CDKN2A methylation is an early change in BE formation” (Line 96-101). The ranges of DNA hypermethylation have been given in Table 1.

  1. Table 1 misses many important genes such as GSTM2, GPX7, MT3, etc.

The total number of new genes added is 8 in addition to the genes you specified.

  1. Page 5 line 120, “CDKN2A has two different upstream exons (1α and 1β) regulated by different promoters. The transcript that was initiated from the proximal promoter (1α) encodes CDKN2A–p16INK4A, while the latter (1β) encodes CDKN2A–p16INK4A [29].” What is the deference between 1a and 1 β? They encode the same protein? 1βencodes CDKN2A–p14ARF?

There was a typo here. It has been corrected to 'the latter (1β) encodes CDKN2A-p14ARF.' Additionally, we have moved this sentence to line 91, which is the paragraph where we initially mentioned CDKN2A, as we believe it is more appropriate there.

  1. Page 5 line 147, “RUNX3 is also found to be hypermethylated in Barrett’s hyperplasia”. What is Barrett’s hyperplasia?

Hyperplasia word has been changed as dysplasia.

Reviewer 2 Report

The presented manuscript is a review of the role of epigenetic alterations in Barrett’s Esophagus to Esophageal Adenocarcinoma. It is an interesting topic. However, I have a few comments.

1.      The Introduction is interesting and well-written.

2.      Genes should be in italics but proteins and enzymes should not be in italics. In addition, authors should better signal when they describe gene expression and when they present protein data. Authors should check and keep them in the correct format throughout the text.

3.      Please expand all abbreviations which appear in the text the first time (e.g. XCL1, XCL3, CCNE1, ZEB-1, ZEB-2).

4.      The weakest feature of the work is only the presentation of data and not their proper commentary. How would the Authors explain such discrepancies in the results of methylation analyses? By what methods were these analyses carried out? (e.g. lines 90-93).

5.      Difficult to understand sentences in lines 113-117. Referring to a protein or gene alone as a biomarker is not appropriate. Only changes in the concentration of this protein or changes in genes (e.g. mutations or SNPs) can be treated as a biomarker.

6.      The Authors do not indicate in the text when they describe p16INK4A and p16INK4A methylation. They often use only the abbreviation CDKN2A (Also known as ARF; MLM; P14; P16; P19; CMM2; INK4; MTS1; TP16; CDK4I; CDKN2; INK4A; MTS-1; P14ARF; P19ARF; P16INK4; P16INK4A; P16-INK4A).

7.      The sentence in lines 141-144 is incomprehensible.

8.      Please provide more information on which mutations have been studied (lines 186-188).

9.      Please provide more information on which oncogenes have been studied. (line 212).

10.   Section ‘Histone Modifications’ requires changes. The title suggests that it contains information about histone modifications. The authors present data on HDAC1 and HDAC2 expression only. The conclusion (‘The examination of histone modifications and chromatin remodeling profiles in BE and EAC may have significant implications as potential biomarkers and targets for therapy’) is not related to the content of the subchapter.

11.   The ‘MicroRNAs’ section does not include recent literature. Better studies on this topic are available (e.g. Int J Mol Sci. 2021 Mar 31;22(7):3640. doi: 10.3390/ijms22073640.; J Clin Med. 2020 Dec 31;10(1):117. doi: 10.3390/jcm10010117.).

12.   Please explain what the sentence ‘enrichment analysis’ means (Table 2).

13.   There is an invalid reference in line 358.

14.   Please provide more information on which miRNAs have been studied (line 387).

15.   Please provide more information on which miRNAs have been studied (line 391).

16.   In my opinion, the authors do not indicate which miRNAs are circulating miRNAs.

17.   Only approx. 26% of references concern the latest reports (from 2018-2023).

Author Response

Dear Reviewer 2,

Thank you for reviewing our article, "Epigenetic Alterations from Barrett’s Esophagus to Esophageal Adenocarcinoma". We appreciate your feedback, and we have revised our work according to your suggestions. We have attached the revised version of our article, along with a point-by-point response to your comments, which are highlighted in yellow. Please see the attached manuscript. 

I also want to inform you that the English language editing of the article was conducted by the Ege University Planning and Monitoring Coordination of Organizational Development and Directorate of Library and Documentation. A thank you statement has been added to the acknowledgement section of the article.

We hope that these revisions address your concerns.

Thank you for your help in improving the quality of our work.

Your comments & the answers:

  1. The Introduction is interesting and well-written.
  2. Genes should be in italics but proteins and enzymes should not be in italics. In addition, authors should better signal when they describe gene expression and when they present protein data. Authors should check and keep them in the correct format throughout the text.

All gene names have been italicized and highlighted in yellow.

  1. Please expand all abbreviations which appear in the text the first time (e.g. XCL1, XCL3, CCNE1, ZEB-1, ZEB-2).

All abbreviations have been written in the text the first time.

  1. The weakest feature of the work is only the presentation of data and not their proper commentary. How would the Authors explain such discrepancies in the results of methylation analyses? By what methods were these analyses carried out? (e.g. lines 90-93).

Methylation ranges were provided separately for each patient group in Table 1. Comments on differences related to methylation were added to line 216. Additionally, commentary on histone modifications was added between lines 303 and 312.

  1. Difficult to understand sentences in lines 113-117. Referring to a protein or gene alone as a biomarker is not appropriate. Only changes in the concentration of this protein or changes in genes (e.g. mutations or SNPs) can be treated as a biomarker.

Other gene names were removed (line 124).

  1. The Authors do not indicate in the text when they describe p16INK4A and p16INK4A methylation. They often use only the abbreviation CDKN2A (Also known as ARF; MLM; P14; P16; P19; CMM2; INK4; MTS1; TP16; CDK4I; CDKN2; INK4A; MTS-1; P14ARF; P19ARF; P16INK4; P16INK4A; P16-INK4A).

"The sentence, 'CDKN2A has two different upstream exons (1α and 1β) regulated by different promoters. The transcript that was initiated from the proximal promoter (1α) encodes CDKN2A–p16INK4A, while the latter (1β) encodes CDKN2A– p14ARF. The majority of CDKN2A studies in BE, HGD, and EAC is CDKN2A–p16INK4A promoter methylation.” has been moved from line 128 to line 91, which is the first occurrence of the CDKN2A gene.

We believe this makes it easier to understand that intense methylation refers to the p16INK4A promoter and that literature on p14ARF methylation is only given in line 129."

  1. The sentence in lines 141-144 is incomprehensible.

The sentence "In a recent study, lesion-containing sections of patients with LGD, HGD, and adenocarcinoma esophageal tissues were separated from histological sections by laser-capture microdissection and p16 (encoded by CDKN2A) methylation analysis with pyrosequencing" was changed to "In a recent study, lesion-containing sections of patients with LGD, HGD, and EAC were separated from histological sections by laser-capture microdissection and p16 (encoded by CDKN2A) methylation analysis were performed with pyrosequencing" on line 145.

  1. Please provide more information on which mutations have been studied (lines 186-188).

The sentence "In epithelial cancers, e-cadherin may be silenced by inherited mutations in LOH, CDH, or promoter hypermethylation in the 16q22;64 chromosome region" was changed to "In epithelial cancers, e-cadherin may be silenced by LOH in 16q22 region, promoter hypermethylation [20] or inherited mutations such as germline large deletions [69] and transcriptional silencing [70] in gastric cancers" (line 191-194). The word "diagnostic" was added to the following sentence (Line 194).

  1. Please provide more information on which oncogenes have been studied. (line 212).

The sentence "Reduced methylation levels in DNA have been linked to an increased expression of oncogenes and a higher likelihood of genome instability [64]" has been revised as "Hypomethylation levels in DNA have been linked to a higher likelihood of genome instability [73] and an increased expression of oncogenes such as Cyclin D1 (CCND1), Cyclin E1 (CCNE1), KRAS, MYC, and Cell division protein kinase 6 (CDK6) [74]." (line 224-227)                

  1. Section ‘Histone Modifications’ requires changes. The title suggests that it contains information about histone modifications. The authors present data on HDAC1 and HDAC2 expression only. The conclusion (‘The examination of histone modifications and chromatin remodeling profiles in BE and EAC may have significant implications as potential biomarkers and targets for therapy’) is not related to the content of the subchapter.

We aimed to write a conclusion that is more compatible with the content at line 303-312.

  1. The ‘MicroRNAs’ section does not include recent literature. Better studies on this topic are available (e.g. Int J Mol Sci. 2021 Mar 31;22(7):3640. doi: 10.3390/ijms22073640.; J Clin Med. 2020 Dec 31;10(1):117. doi: 10.3390/jcm10010117.).

We would like to inform you that we have included both publications, Zarrilli et al. and Inokuchi et al., in our reference list. Zarrilli et al.'s study had a broad interest in esophageal cancers, and we have found that many of the references they used for EAC are also cited in our reference list.

  1. Please explain what the sentence ‘enrichment analysis’ means (Table 2).

The explanation about the enrichment analysis has been appended to the Table 2.

  1. There is an invalid reference in line 358.

We edited the invalid references (line 390).

  1. Please provide more information on which miRNAs have been studied (line 387).

We have been provide more information about the mentioned study, at line 417-431.

  1. Please provide more information on which miRNAs have been studied (line 391).

We have been provide more information about the mentioned study, at line 433-451.

  1. In my opinion, the authors do not indicate which miRNAs are circulating miRNAs.

The majority of miRNAs present in circulation have also been observed in tissues, hence we have refrained from creating a distinct classification. The primary objective of our inclusion of examples of circulating miRNAs was not to establish definitive categories, but instead to highlight recent scientific findings in this area.

  1. Only approx. 26% of references concern the latest reports (from 2018-2023).

The methylation, histone modification, and miRNA studies in the review were presented by scanning through the years, which led to a high number of older studies, especially in hypermethylation. As a result, the latest report percentages may appear low. However, every study has been examined up to the present day. To decrease this percentage, the old references in the introduction section have been replaced with newer ones.

Round 2

Reviewer 2 Report

The Authors have addressed all my concerns.